Predicting hotel booking cancellations using tree-based neural network

Yang Dan hyh55100118@163.com
Miao Xiaoling
Wuhan Polytechnic , Wuhan City, Hubei Province , China
Nguyen Hoang
Electronic publication date: 2024 Nov 18
Publication date: 2024
Volume: 10
Electronic Location ID: e2473
Received 2024 Apr 10; Accepted 2024 Oct 11
Copyright: © 2024 Yang and Miao
Copyright year: 2024
Copyright holder: Yang and Miao
License: This is an open access article distributed under the terms of the Creative Commons Attribution License, which permits unrestricted use, distribution, reproduction and adaptation in any medium and for any purpose provided that it is properly attributed. For attribution, the original author(s), title, publication source (PeerJ Computer Science) and either DOI or URL of the article must be cited.
License URL: https://creativecommons.org/licenses/by/4.0/

Keywords: Data science, Machine learning, Tree-based neural network, Booking prediction, Hotel

Funding: The authors received no funding for this work.

==============================
In the hospitality business, cancellations negatively affect the precise estimation of revenue management. With today’s powerful computational advances, it is feasible to develop a model to predict cancellations to reduce the risks for business owners. Although these models have not yet been tested in real-world conditions, several prototypes were developed and deployed in two hotels. The their main goal was to study how these models could be incorporated into a decision support system and to assess their influence on demand-management decisions. In our study, we introduce a tree-based neural network (TNN) that combines a tree-based learning algorithm with a feed-forward neural network as a computational method for predicting hotel booking cancellation. Experimental results indicated that the TNN model significantly improved the predictive power on two benchmark datasets compared to tree-based models and baseline artificial neural networks alone. Also, the preliminary success of our study confirmed that tree-based neural networks are promising in dealing with tabular data.

Introduction

Revenue management is defined as a variety of activities to allocate the ‘right’ capacity to the ‘right’ customer with the ‘right’ cost in the ‘right’ time’ (Kimes & Wirtz, 2003; Chiang, Chen & Xu, 2007; Ban et al., 2023). Since then, this concept has been widely applied in service industries, such as hospitality, rental, and entertainment services (Kimes & Wirtz, 2003; Leow, Nguyen & Chua, 2021; Luo, Zhuo & Xu, 2023). In the hospitality industry, a good service provider manages to make the ‘right’ accommodation available for the ‘right’ guest with the ‘right’ cost at the ‘right’ time via the ‘right’ distribution channel (Hayes, Hayes & Hayes, 2021). Traditionally, hotels accommodate potential guests by allowing prior reservations. A confirmed reservation is a legal agreement between the hotel and the guest(s) to provide the booked service(s) at a certain future date and for the agreed-upon fee. However, most contracts permit early cancellation before the supply of services so that it shifts the risk onto the hotel, as it is required to ensure the availability of rooms for guests who arrive as scheduled, as well as bear the cost of unoccupied rooms in the event of a cancellation or no-show (Talluri & Ryzin, 2004). As a result, cancellations have a major effect on demand-management choices made within the scope of revenue management (Talluri & Ryzin, 2004). Precise prediction, one of the key components of an effective revenue management plan, is significantly influenced cancellations. Specifically, it has been found that hotel cancellations may account for up to 20% of total reservations and up to 60% in the case of airport/roadside hotels (Morales & Wang, 2010). In an effort to mitigate their losses, hotels adopt overbooking techniques and stringent cancellation policies (Talluri & Ryzin, 2004; Smith et al., 2015). Nonetheless, these demand-management strategies might have an adverse effect on the hotel’s earnings and social image. Overbooking may lead a hotel to reject service to a client, which negatively affects the business’s relationship with the guest and incurs fees for relocating the guest to an other hotel (Noone & Lee, 2011). The hotel may lose future bookings since the relocation can introduce the guest to a preferable hotel. Once the considerable price reductions are applied, restrictive cancellation policies (e.g., non-refundable policies or policies with 48-h prior cancellation deadlines) reduce both income and the number of reservations (Chen, Schwartz & Vargas, 2011; Smith et al., 2015).

Several studies investigated solutions to mitigate the effects of cancellations on revenue, inventory allocation, and overbooking strategies, cancellation rules (Talluri & Ryzin, 2004; Ivanov, 2014). Most published works focus on aviation which differs from hospitality in some aspects (Freisleben & Gleichmann, 1993; Subramanian, Stidham & Lautenbacher, 1999; Hueglin & Vannotti, 2001; Garrow & Ferguson, 2008; Lemke, 2010; Yoon, Lee & Song, 2012). However, the number of studies related to the hotel business has been expanding, indicating the topic’s significance for this industry (Weatherford & Kimes, 2003; Caicedo-Torres & Payares, 2016; Schwartz et al., 2016). While numerous research employed conventional statistical learning methods, a few studies used machine learning approaches (Freisleben & Gleichmann, 1993; Hueglin & Vannotti, 2001; Caicedo-Torres & Payares, 2016). Among numerous published methods for predicting cancellations, there are only four studies that addressed this issue in hotel business (Liu, 2004; Morales & Wang, 2010; Huang, Chang & Ho, 2013; Antonio, de Almeida & Nunes, 2017) and three of them used information from property management systems (PMS) (Liu, 2004; Antonio, de Almeida & Nunes, 2017; António, de Almeida & Nunes, 2017). The most recent research, however, incorporates Passenger Name Record (PNR) data, the standard data for the airline industry sourced from the International Air Transport Association. Only a few studies have used a classification issue as opposed to the regression problem adopted by the vast majority of studies on booking cancellations (Antonio, de Almeida & Nunes, 2017; António, de Almeida & Nunes, 2017). While the former works concentrate on anticipating worldwide cancellation rates, the latter works are more concerned with estimating the chance of cancellation for individual bookings. Morales & Wang (2010) supposed that precisely estimating whether a reservation would be canceled or not is challenging. In contrast, Antonio, de Almeida & Nunes (2017), António, de Almeida & Nunes (2017) demonstrated that predicting the cancellation likelihood of any bookings was feasible. The hotel net demand may be calculated by subtracting all expected cancellations from the demand. A hotel’s revenue manager can make better demand-management choices and enhance overbooking and cancellation policies with precise demand data. Sánchez-Medina & C-Sánchez (2020) implemented multiple models, including Random Forests, Support Vector Machines, C.5.0, and an artificial neural network optimized by genetic programming, to predict hotel booking cancellations. Similarly, Sánchez, Sánchez-Medina & Pellejero (2020) developed an ensemble of three models: support vector machines, C.5.0, and artificial neural networks, for the same purpose. Most recently, Herrera et al. (2023) utilized a variety of models, such as multilayer perceptron neural networks, radial basis function neural networks, decision trees, Random Forests, AdaBoost, and XGBoost, to further enhance the performance of cancellation prediction in hotel booking.

Recent deep learning applications have demonstrated strong performance across various problems, particularly in handling unstructured data like images, text, videos, audio, and more (Mobahi, Collobert & Weston, 2009; LeCun, Bengio & Hinton, 2015; He et al., 2015; Devlin et al., 2018; Le Nguyen Quoc et al., 2019). Deep learning has not been proven as a promising solution for tabular data comprising continuous and discrete features. Traditional machine learning, especially tree-based approaches, is usually employed to deal with tabular data. All tree-based methods adopt ensemble learning strategies based on decision trees so that they tend to perform better on this type of data (Chui et al., 2018). Despite the prevalence of tabular data in the real world, deep learning has not been widely used for tabular data. Deep neural networks utilize a nonlinear function to map the link between input characteristics and target vectors, while decision trees use a piecewise function to categorize data and attain a certain objective. Tree-based models provide a more interpretable piecewise function than neural networks, but they are unable to capture intricate interactions. Neural networks, on the other hand, are more effective at modeling complex relationships, but they are less interpretable. However, when the data is great and high-dimensional, modeling using tree-based algorithms is challenging. An idea of deep neural networks incorporated with tree-based learning algorithms has been introduced, designed, and released with an expectation to take advantages of both deep neural networks and tree-based approaches (Sarkar, 2022). In an extension of this implementation, more tree-based algorithms and different neural network designs can be combined to improve the performance of the prediction model or make it more adaptable to specific datasets.

In this study, we propose a tree-based neural network (TNN) integrating a tree-based learning algorithm into a feed-forward neural network as a novel approach to predict hotel booking cancellation. The TNN was inspired by the work of Sarkar (2022) with modifications. The key difference between Sarkar’s (2022) work and ours lies in the focus. While Sarkar (2022) proposed a novel method called XBNet, their work does not provide specific guidance on the optimal number of layers or the types of tree-based algorithms that should be integrated with neural networks. In contrast, our research emphasizes the practical implementation of a range of tree-based algorithms combined with various neural network architectures to identify the best predictive model. We acknowledge the valuable contribution of Sarkar (2022), which serves as a solid foundation for further studies, including ours, that aim to expand upon their technical ideas and provide more detailed insights into the optimal model configurations for enhanced performance. The TNN model is trained with the optimization technique called Boosted Gradient Descent, which is initialized with the feature importance of a gradient-boosted tree, and it updates the weights of each neural network layer. The TNN model is benchmarked with traditional tree-based models and state-of-the art neural networks to observe the improved performance of our proposed method.

Materials and Methods

Datasets

We used the dataset provided by Antonio, de Almeida & Nunes (2019) for our study. They collected the dataset from a hotel chain in Portugal. For privacy and ethical reason, the name of the hotel chain was kept unexposed. The hotel chain consented to join and authorize the data collectors to access the Property Management System (PMS) data of their two accommodations. The first accommodation, termed H1, is a resort while the second one, termed H2, is a city hotel. Both accommodations are certified with four-stars and have more than 200 rooms. Data for the experiment were recorded during the period between July 2015 and August 2017. The PMS of the hotel chain has confirmed that its database tables do not contain any missing data. However, certain categorical variables, such as “Agent” and “Company”, include “NULL” as one of the categories. The “NULL” in this case is interpreted as “not applicable”, not “missing value”. The “Agent” field of a booking is blank, for instance means that the reservation was not made through a travel agent. The datasets H1 and H2 contain 40,060 and 79,330 booking samples with the same number of features (variables). For each original dataset, there are 31 variables, including 14 categorical variables, 16 numeric variables, and 1 date variable. The variables ‘AssignedRoomType’, ‘RequiredCarParkingSpaces’, and ‘ReservedRoomType’ were removed by using feature importance ranking. The ‘country’ variable was also taken out of the modeling because it caused information leakage since this information was only confirmed and corrected at check-in. Hence, Portugal was considered the country of origin when booking a hotel. Song & Li’s (2008) study found that the phenomenon of seasonality plays a crucial role in the tourism industry which affected the canceled booking ratio. Therefore, all variables related to time or one representing seasonality, including ‘ArrivalDateDayOfMonth’ (date), ‘ArrivalDateMonth’ (month), ‘ArrivalDateWeekNumber’ (yearly week number), ‘ArrivalDateYear’ (year), ‘ReservationStatusDate’ (status date of reservation), were eliminated from the modeling. Table 1 provides information of refined datasets used for model development and evaluation.

Table 1 Datasets for model development and evaluation.

Dataset	Training data	Test data	
	No. of samples	Start date	End date	No. of samples	Start date	End date	
H1	30,045 (75%)	01/07/2015	04/03/2017	10,015 (25%)	05/03/2017	31/08/2017	
H2	59,498 (75%)	01/07/2015	24/03/2017	19,832 (25%)	25/03/2017	31/08/2017	

Figure 1 shows the annual cancellation rate in both accommodations. During the period, the cancellation ratios of resort H1 were unstable with wide fluctuations in the range from 15% to 40%. Similarly, after having a high cancellation ratio at the beginning of July and August 2015, the cancellation ratios of hotel H2 oscillated between 25% and 50%. In addition, while the overall cancellation ratio of resort H1 was lower than that of hotel H2, its trending line moderately increased. In contrast, the trending line of the hotel H2 slightly decreased. Figure 2 displays the annual cancellation rate in both accommodations. The figure indicates that seasonality played a crucial role in the cancellations behaviors in that period. The cancellation rate dropped to a minimum in the off-peak season months (November and January) and raised to a maximum from June to September.

Figure 1 The monthly change in cancellation rate of both accommodations.

Figure 2 Annual cancellation rate in both accommodations.

Overview of the method

Figure 3 describes the main steps in developing our tree-based neural network model. Initially, original datasets were split into training sets (75%) and test sets (25%) using the time series cross-validator. The splitter function generates indices for dividing a dataset of time series data, which is collected at fixed intervals, into two subsets: a training set and a test set. It is important to note that, in each iteration of the splitting process, the test indices must be sequentially higher than those used in the previous split. As such, randomly shuffling the data during cross-validation is not appropriate in this context. In contrast to standard cross-validation techniques, the training sets in successive splits are supersets of the preceding training sets. Time series cross-validation was conducted on the training set to find the best-performing tree-based model. The selected tree-based model was then integrated into an artificial neural network (ANN) to create a tree-based neural network (TNN). Two training sets were used to develop the two TNN models (for H1 and H2). These models were evaluated using two corresponding test sets.

Figure 3 Main steps in developing tree-based neural network model.

Tree-based learning algorithms

We constructed four models using four tree-based algorithms, including gradient boosting, extreme gradient boosting, Random Forest, and Extremely Randomized Trees, and examined these models using cross-validation to find the best-performing tree-based model which was then integrated into the deep neural network.

Gradient boosting

Gradient boosting (GB) (Friedman, 2001, 2002) is a type of supervised ensemble learning in which a weak learner is combined with the efforts of other learners in order to improve its performance. The process involves training new weak learners to handle the more difficult data by selecting only those observations that the weak learner is able to handle and discarding the rest. The use of freely differentiable loss functions allows GB to be adapted for various applications, including regression, multi-class classification, and other tasks.

Extreme gradient boosting

Extreme gradient boosting (XGB) (Chen & Guestrin, 2016) is a novel supervised tree-based learning method in which the principle of Classification and Regression Trees (CART) (Steinberg & Colla, 2009) and Gradient Tree Boosting (GTB) (Mason et al., 2000; Friedman, 2001) are integrated. For regularization, it has L1 and L2 penalization options. In order to optimize its performance, XGB is designed to minimize a regularized objective function that combines a convex loss function with a penalty scoring function based on the discrepancy between the predicted and true labels. In each iteration of its boosting, random subsets of data and features are used, and the weight of incorrectly predicted classes is raised. Its robustness was confirmed by many studies targeting diverse problems.

Random Forest

Random Forest (RF) (Breiman, 2001) is a supervised ensemble learning method that utilizes the “bagging” principle (Breiman, 1996) and random feature selection (Ho, 1995) to create a robust and effective predictor. It generates multiple unique decision trees and combines the outputs to address classification and regression problems. The outputs are modified based on either the mode of the classes or the average of the predicted values from the various trees. One notable advantage of RF is its ability to overcome the “overfitting tendency” of decision tree algorithm, which is a common weakness of the tree-based family.

Extremely randomized trees

Extremely Randomized Trees (ERT) (Geurts, Ernst & Wehenkel, 2006) is a supervised ensemble learning method involving significantly randomizing the selection of variables and cut points used to split tree nodes, and in some cases, the generated trees have no relation to the output values of the training sample. The intensity of the randomization can be adjusted to meet specific requirements. ERT is dominant over other tree-based methods in terms of computing speed while its performance is still highly competitive. Like other robust tree-based approaches, it can be used to tackle classification and regression problems.

Tree-based neural network

Tree-based neural network (TNN) is a novel deep learning approach that combines the advantages of a tree-based learning algorithm and a deep neural network (DNN) to improve the performance of the conventional DNN in dealing with tubular data. Any tree-based learning algorithm can be integrated into DNN to create a TNN model. The detailed principle of TNN is clearly explained in Sarkar’s (2022) study. The TNN is designed with n gradient-boosted layers in which the fully-connected (FC) layer uses feature importance computed by the tree-based algorithm as the neural network’s weights (Fig. 4). In each gradient-boosted layer, the flow of data follows three consecutive steps: (S1)–Training batch tree-based model, (S2)–Exporting batch feature importances from trained tree-based model and using them as weights of FC layer, and (S3)–Training FC layer. The gth gradient-boosted layer adopts outputs from (g−1)th layer as inputs. In the first gradient-boosted layer, the FC layer is initialized without weight and it uses computed feature importance as its weights. From the second gradient-boosted layer, the FC layer is initialized with random weights and it iteratively updates its weight by adding the scaled batch feature importance into its current weights. Except for the final gradient-boosted layer using sigmoid as its activation function, other gradient-boosted layers are activated by LeakyReLU. In this study, we used the learning rate of 1 × 10−4 and the minibatch of 128. To develop the TNN models, we used three simple DNN: (TNN1) Dense(32) ← Dense(32) ← Dense(1), (TNN2) Dense(64) ← Dense(32) ← Dense(1), and (TNN3) Dense(128) ← Dense(64) ← Dense(1).

Figure 4 Tree-based neural network architecture.

In the context of this study, three different designs of this model architectures are proposed. The numbers of parameters of fully connected layers 1, 2, and 3 in these three designs are (32, 32, 1), (64, 32, 1), and (128, 64, 1), respectively. These three designs are presented as (TNN1) Dense(32) ← Dense(32) ← Dense(1), (TNN2) Dense(64) ← Dense(32) ← Dense(1), and (TNN3) Dense(128) ← Dense(64) ← Dense(1).

Results and discussion

Evaluation metrics

To evaluate the performance of the model, several metrics, includingspecificity (SP), recall (RE), balanced accuracy (BA), precision (PR), F1 score (F1), Matthews’s correlation coefficient (MCC), the area under the receiver operating characteristic curve (AUCROC), and the area under the precision-recall curve (AUCPR), were employed. These metrics were calculated using the True Positive (TP), False Positive (FP), True Negative (TN), and False Negative (FN) values.

Tree-base model selection

Table 2 gives information on the time series cross-validation performance of the four tree-based models. For dataset H1, the GB model is the best predictor, followed by the ERT model, RF model, and XGB models with the cross-validation AUCROC values of 0.7691, 0.7676, 0.7619, and 0.7589, respectively. For dataset H2, the RF model works more effectively compared to GB, XGB, and ERT models with corresponding cross-validation AUCROC values of 0.8082, 0.8016, 0.7914, and 0.7909, respectively. Based on the results, the GB model was selected as the tree-based model to be integrated into DNN for datasets H1 while the RF model was selected to be integrated into DNN for datasets H2.

Table 2 Time series cross-validation performance of the four tree-based models.

Model	AUCROC	
	H1	H2	
Gradient boosting	0.7691	0.8016	
Extreme gradient boost	0.7589	0.7914	
Random forest	0.7619	0.8082	
Extremely randomized tree	0.7676	0.7909	

Model evaluation

Table 3 provides results on the performance of the three TNN models compared to the four tree-based models (as baseline models) on the test set. The results show that all TNN models have better performance compared to all tree-based models in terms of AUCROC and AUCPR values. Among these TNN models, the TNN1 model is the best-performing model for dataset H1 while the TNN2 model is dominant over the others for dataset H2. For the dataset H1, the TNN1 model achieves an AUCROC value of about 0.86 and an AUCPR value of about 0.73, followed by the TNN2 and TNN3 models. For the dataset H2, the TNN3 model works more efficiently than the TNN2 and TNN3 models. From these results, there are two opposite trends found in which the simpler TNN model works better in dataset H1 while the more complex TNN model obtains higher performance in dataset H2. The difference in the number of training data may be one of the possible reasons explaining this observation.

Table 3 The performance of the TNN models compared to baseline tree-based models on the test set.

Dataset	Model	Metric	Time (s)	
		AUCROC	AUCPR	BA	SN/RE	SP	PR	MCC	F1		
H1	TNN1	0.8554	0.7335	0.6089	0.9950	0.2228	0.3743	0.2815	0.5439	6.86	
	TNN2	0.8399	0.7132	0.6185	0.9956	0.2414	0.3801	0.2968	0.5502	7.26	
	TNN3	0.8377	0.7059	0.6227	0.9950	0.2504	0.3828	0.3027	0.5528	8.58	
	GB	0.7643	0.6214	0.5530	0.1104	0.9956	0.9215	0.2578	0.1971	2.17	
	XGB	0.7718	0.6242	0.5572	0.1207	0.9937	0.8995	0.2636	0.2129	3.00	
	RF	0.7639	0.6127	0.5340	0.0715	0.9965	0.9048	0.2022	0.1325	3.56	
	ERT	0.7434	0.5815	0.5049	0.0103	0.9994	0.8919	0.0750	0.0205	3.12	
H2	TNN1	0.8307	0.8189	0.5525	0.9915	0.1134	0.4510	0.2047	0.6200	13.23	
	TNN2	0.8370	0.8130	0.5583	0.9932	0.1233	0.4542	0.2200	0.6233	15.99	
	TNN3	0.8272	0.7586	0.5612	0.9942	0.1283	0.4558	0.2277	0.6251	18.13	
	GB	0.7760	0.6888	0.7148	0.6106	0.8190	0.7125	0.4415	0.6576	4.01	
	XGB	0.7812	0.7010	0.7115	0.6092	0.8138	0.7061	0.4341	0.6541	5.78	
	RF	0.7837	0.7041	0.7123	0.6022	0.8225	0.7136	0.4378	0.6531	7.12	
	ERT	0.7484	0.6707	0.6983	0.5978	0.7988	0.6857	0.4060	0.6387	8.26	

Model benchmarking

Table 4 summarizes the results on the performance of the three TNN models compared to the four state-of-the-art models on the test set. These four algorithms include RB-NN (Li & Micchelli, 2000), DNN (Mittal, 2020), XBNet (Sarkar, 2022), Multilayer Perceptron (MLP) (Singh & Banerjee, 2019), and Bayesian Networks incorporated with Lasso regression (BN-Lasso) (Chen et al., 2023). The results reveal that all TNN models outperform the other models in terms of AUCROC. However, for AUCPR, TNN3 performs less effectively compared to the DNN and XBNet models. On the H1 and H2 datasets, TNN1 and TNN2 remain the best-performing models, with AUCROC values of about 0.86 and 0.84, respectively. The TNN1 model also outperforms the other models, as well as the two TNN designs, with AUCPR values of about 0.73 and 0.82, respectively. The XBNet and DNN models are competitive, as their performance is not significantly different from ours. The RB-NN models show the fastest training speed, indicating the most cost-effective methods. Our proposed TNN models take longer to complete one training epoch compared to the other models.

Table 4 The performance of the TNN models compared to state-of-the-art models on the test set.

Dataset	Model	Metric	Time (s)	
		AUCROC	AUCPR	BA	SN/RE	SP	PR	MCC	F1		
H1	TNN1	0.8554	0.7335	0.6089	0.9950	0.2228	0.3743	0.2815	0.5439	6.86	
	TNN2	0.8399	0.7132	0.6185	0.9956	0.2414	0.3801	0.2968	0.5502	7.26	
	TNN3	0.8377	0.7059	0.6227	0.9950	0.2504	0.3828	0.3027	0.5528	8.58	
	RB-NN	0.7533	0.5440	0.6615	0.3411	0.4851	0.8380	0.5831	0.5296	6.13	
	DNN	0.8328	0.6777	0.7530	0.4821	0.7400	0.7659	0.5963	0.6604	8.96	
	XBNet	0.8439	0.7173	0.7480	0.4662	0.9047	0.5913	0.5084	0.6510	6.79	
	MLP	0.7710	0.6064	0.6874	0.3521	0.8191	0.5558	0.4628	0.5914	11.26	
	BN-Lasso	0.7895	0.6170	0.6795	0.3893	0.4886	0.8703	0.6377	0.5533	13.25	
H2	TNN1	0.8307	0.8189	0.5525	0.9915	0.1134	0.4510	0.2047	0.6200	13.23	
	TNN2	0.8370	0.8130	0.5583	0.9932	0.1233	0.4542	0.2200	0.6233	15.99	
	TNN3	0.8272	0.7586	0.5612	0.9942	0.1283	0.4558	0.2277	0.6251	18.13	
	RB-NN	0.7695	0.6863	0.7112	0.4209	0.6786	0.7437	0.6604	0.6694	10.24	
	DNN	0.8223	0.8121	0.7235	0.4428	0.7934	0.6536	0.6272	0.7006	13.12	
	XBNet	0.8175	0.8086	0.7111	0.4209	0.8091	0.6131	0.6057	0.6928	12.86	
	MLP	0.7458	0.6393	0.6998	0.3973	0.6754	0.7242	0.6426	0.6586	17.95	
	BN-Lasso	0.7943	0.7433	0.7191	0.4406	0.6611	0.7772	0.6854	0.673	19.11	

Limitations and future directions

Although combining the strengths of tree-based models and neural networks can enhance interpretability and performance through a unique optimization technique for tabular data, tree-based neural networks like ours, have several limitations besides their innovation and robustness. The combination of tree-based algorithms (e.g., Random Forest, eXtreme gradient boosting) and neural networks increases model complexity and computational intensity. It also raises the risk of overfitting, especially with smaller datasets, requiring careful regularization. Hyperparameter tuning is complex, as both models have numerous settings that need optimization. Data preprocessing can be challenging due to the differing requirements of tree-based algorithms and neural networks. Additionally, tree-based algorithms may not generalize well to unstructured data, and their performance gains over simpler models may be marginal, making the added complexity sometimes unjustified. Finally, there is limited theoretical understanding of how these models interact, leaving some aspects of their behavior underexplored.

To address these limitations, future directions could explore the use of data-centric AI approaches (Wang et al., 2024). Instead of focusing solely on refining the model architecture, data-centric AI emphasizes improving data quality, consistency, and relevance, which could mitigate overfitting and enhance generalization, particularly on smaller datasets (Hussain et al., 2020; Tran & Nguyen, 2024). This approach could streamline preprocessing by standardizing and optimizing data for both tree-based algorithms and neural networks (Wang, Chukova & Nguyen, 2023). Additionally, focusing on creating richer, more representative datasets may reduce the need for extensive hyperparameter tuning and complex model combinations, ultimately simplifying the pipeline while maintaining strong performance.

Conclusion

Since the accurate estimation of revenue management is negatively influenced by canceled bookings, investigating a method to estimate the booking cancellation is essential. In this study, we proposed the tree-based neural network model, a novel neural network architecture in which a tree-based learning algorithm is combined with a feedforward neural network. The integrated tree-based model learns the tubular features and exports the feature importances, which are then used as the neural network’s weights to improve its efficiency in processing tubular data. The experimental results confirmed the robustness of our proposed method compared to all baseline tree-based models. The use of this novel architecture can be extended to diverse problems whose data structures are tubular.

Supplemental Information

Supplemental Information 1 Python code used in the study.

Additional Information and Declarations

Competing Interests

Author Contributions

Data Availability

The authors declare that they have no competing interests.

Dan Yang conceived and designed the experiments, performed the experiments, analyzed the data, performed the computation work, prepared figures and/or tables, authored or reviewed drafts of the article, and approved the final draft.

Xiaoling Miao conceived and designed the experiments, performed the experiments, analyzed the data, performed the computation work, prepared figures and/or tables, authored or reviewed drafts of the article, and approved the final draft.

The following information was supplied regarding data availability:

The Python code used in this study is available in the Supplemental Files.

The data is available at https://doi.org/10.1016/j.dib.2018.11.126.

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
