# Peer review of "Predicting hotel booking cancellations using tree-based neural network"

_PeerJ Computer Science, doi:10.7717/peerj-cs.2473_

## Round 0.1 · original submission · Major Revisions

According to reviewers' comments, your manuscript is not ready for publication in its current form due to various concerns. However, we are pleased to give you an offer to revise it based on reviewers' comments.

Reviewer 1 ·

Basic reporting

No comment

Experimental design

• Problem novelty: the idea of using TNN for forecasting hotel booking cancellations is not new and there are many works have been done it, such as Sarkar 2022 [1] in the manuscript or recent works [2-4].
• Technical novelty and soundness: The authors should clearly state the difference between their proposed method and Sarkar 2022 [1] to support their claim about “introduce a TNN”.

[1] Sarkar, T. (2022). Xbnet: An extremely boosted neural network. Intelligent Systems with Applications, 15:200097.
[2] Herrera, A., Arroyo, Á., Jiménez, A., & Herrero, Á. (2024). Forecasting hotel cancellations through machine learning. Expert Systems, e13608.
[3] Sánchez-Medina, Agustín & C-Sánchez, Eleazar. (2020). Using machine learning and big data for efficient forecasting of hotel booking cancellations. International Journal of Hospitality Management. 89. 102546. 10.1016/j.ijhm.2020.102546.
[4] C-Sánchez, Eleazar & Sánchez-Medina, Agustín & Pellejero, Mónica. (2020). Identifying critical hotel cancellations using artificial intelligence. Tourism Management Perspectives. 35. 100718. 10.1016/j.tmp.2020.100718.

Validity of the findings

• Experiment:
o The authors should show more details about three variations of TNN (TNN1, TNN2, TNN3), what is the difference between them?
o The author should compare with [1] because the proposed method is based on that method.
o The author should compare with other ANN methods, such as MLP [5], RBF neural networks [6], DNN [7].
• Writing and presentation: The paper is clear and easy to understand.

[5] Singh, J., & Banerjee, R. (2019). A study on single and multi-layer perceptron neural network. In 2019 3rd international conference on computing methodologies and communication (ICCMC), pp. 35–40.
[6] Li, X., & Micchelli, C. A. (2000). Approximation by radial bases and neural networks. Numerical Algorithms, 25(1), 241–262.
[7] Mittal, S. (2020). A survey on modeling and improving reliability of DNN algorithms and accelerators. Journal of Systems Architecture, 104, 101689.

Cite this review as

·

Basic reporting

The study aims to design a prediction model to forecast hotel booking cancellations using tree-based neural networks. The manuscript was presented in good form for review with good English and organization. The work has been done on two different datasets to demonstrate the effectiveness of their proposed method. The literature review on related works is focused. This work is interesting and has achieved very good outcomes, but there are major and minor points that need to be revised.

Experimental design

Major points:
(i) As your proposed method is a combination of neural networks and tree-based machine learning algorithms, comparing your method to only tree-based algorithms is not enough. I suggest the authors implement several neural network models for a fair comparison.
(ii) Is there any reason for the proposed architecture? How many fully connected (FC) layer should be used? Please clarify this point.
(iii) Limitations of your work need to be discussed besides your achievements.
(iv) Combining neural networks and tree-based algorithms seem to require higher computional cost. Authors to provide comparative table for computational cost among these implemented models.
Minor points:
(i) If your propoposed method is inspired from Sarkar’s study, please cite his/her work in the Introduction work.
(ii) Section “RESULTS AND DISCUSSIONS” should be “RESULTS AND DISCUSSION”
(iii) Section “CONCLUSIONS” should be “CONCLUSION”.

Validity of the findings

The work needs to improve the validity of its findings by providing statistical evidence on how their method’s performance fluctuates across multiple trials. Since this dataset contains time-series samples, the data sampling needs to be carefully done with a clear explanation added to the main text.

---

## Round 0.2 · Minor Revisions

Thank you for submitting your revised manuscript, entitled 'Predicting Hotel Booking Cancellations Using Tree-Based Neural Networks,' to PeerJ Computer Science. While the authors have made a commendable effort in addressing the reviewers' concerns, there are still some issues raised by the reviewers that need to be addressed. The authors are required to revise the manuscript and respond to each of the reviewers' comments individually.

Reviewer 1 ·

Basic reporting

About my concern on problem novelty, the authors have revised the manuscript with more discussions based on my suggestions. However, the second part of my suggestion is that the problem or topic is not new as some recent works [2-4] have been explored. This one is not critical point. So, if possible, I would recommend discussing about these recent works in the manuscript.

Experimental design

Please add the difference between the proposed method and Sarkar 2022 to the manuscript. So that, the readers can understand the proposed method better.

Validity of the findings

The authors have addressed my concern on the three variations of TNN. Please also add the explanation in the revision to the final version of the manuscript, because the revised caption of Figure 4 is not clear as the explanation in the revision.

For experiment part, the authors have conducted additional experiments on other ANN methods. The proposed method achieved better performance in certain metrics and comparable processing time. The proposed method can add value by offering a different approach to community.

Additional comments

N/A

Cite this review as

·

Basic reporting

The manuscript is well-organized and has clear English, but some sentences could be simplified for better readability. The following sentences could be shortened or split into two sentences to improve clarity:

- "Although these kinds of models have never been tested in real-world conditions, several prototypes were constructed and deployed in two hotels to study how these models may be incorporated into a decision support system and their influence on demand-management choices". (lines 11 - 14)

- "Due to the implementation of considerable price reductions, restrictive cancellation policies (e.g., non-refundable policies or policies with 48-hour prior cancellation deadlines) reduce both income and the number of reservations." (lines 42 - 44)

- "The integrated tree-based model learns the tubular features and exports the feature importances to be used as the neural network’s weight to improve the efficiency of the neural network when dealing with tubular data." (lines 256 - 258)

Experimental design

The experiments were appropriately conducted. Authors have addressed all my concerns. Results of additional experiments demonstrate the model robustness.

Validity of the findings

Statistical evidence is fully provided to confirm the reproducibility of this work.

Reviewer 3 ·

Basic reporting

1. The proposed method applies a tree-based neural network for Predicting hotel booking cancellations. The method section mainly describes 1) the four tree-based algorithms which are very popular, 2) the Tree-based Neural Network introduced in Sarkar, 2022. What are the contributions of this work?

2. The proposed method needs to prove the effectiveness of the simple fully-connected (FC) layer compared to other modern techniques such as CNNs, transformer architectures.

3. The method need to compare with recent works (3 years), such as https://doi.org/10.1016/j.dss.2023.113959

Experimental design

4. The manuscript should conduct some ablation studies.

Validity of the findings

5. The novelty of this work should be considered.

Additional comments

6. The manuscript needs significant improvement.

Cite this review as

---

## Round 0.3 · accepted · Accept

Thank you for your revised work, entitled "Predicting hotel booking cancellations using tree-based neural network", in PeerJ Computer Science. Authors have shown great effort in addressing reviewer's concerns about this work.

·

Basic reporting

- The revised version now meets the requirements for publication in this journal and has addressed all of my concerns.

Experimental design

- The experimental design is well explained and reflects the objectives of the study.
- The additional experiments supported the conclusion of this work.

Validity of the findings

- The findings of this work are valid. The conclusion is clear and indicates that the work achieved its proposed objectives.

Reviewer 3 ·

Basic reporting

The authors have solved the issues that I have discussed.

Experimental design

The authors have solved the issues that I have discussed.

Validity of the findings

The authors have solved the issues that I have discussed.

Cite this review as